

# Adaptive evolution at mRNA editing sites in soft-bodied cephalopods

Mikhail Moldovan[1], Zoe Chervontseva[1,2], Georgii Bazykin[1,2] and Mikhail S. Gelfand[1,2]

[1] Skolkovo Institute of Science and Technology, Moscow, Russian Federation
[2] A.A.Kharkevich Institute for Information Transmission Problems (RAS), Moscow, Russian Federation

## ABSTRACT

**Background:** The bulk of variability in mRNA sequence arises due to mutation—change in DNA sequence which is heritable if it occurs in the germline. However, variation in mRNA can also be achieved by post-transcriptional modification including mRNA editing, changes in mRNA nucleotide sequence that mimic the effect of mutations. Such modifications are not inherited directly; however, as the processes affecting them are encoded in the genome, they have a heritable component, and therefore can be shaped by selection. In soft-bodied cephalopods, adenine-to-inosine RNA editing is very frequent, and much of it occurs at nonsynonymous sites, affecting the sequence of the encoded protein.
**Methods:** We study selection regimes at coleoid A-to-I editing sites, estimate the prevalence of positive selection, and analyze interdependencies between the editing level and contextual characteristics of editing site.
**Results:** Here, we show that mRNA editing of individual nonsynonymous sites in cephalopods originates in evolution through substitutions at regions adjacent to these sites. As such substitutions mimic the effect of the substitution at the edited site itself, we hypothesize that they are favored by selection if the inosine is selectively advantageous to adenine at the edited position. Consistent with this hypothesis, we show that edited adenines are more frequently substituted with guanine, an informational analog of inosine, in the course of evolution than their unedited counterparts, and for heavily edited adenines, these transitions are favored by positive selection. Our study shows that coleoid editing sites may enhance adaptation, which, together with recent observations on *Drosophila* and human editing sites, points at a general role of RNA editing in the molecular evolution of metazoans.

Subjects Bioinformatics, Evolutionary Studies, Genetics, Genomics, Zoology
Keywords RNA editing, Cephalopods, Coleoids, Adaptation, Selection, Positive selection, Genome evolution, Polymorphism, Evolvability

# INTRODUCTION

The process of natural selection requires heritable variation to be present in a population and the absence of genetic variants selection could act upon is generally considered to be a factor hampering adaptation (*Lush, 1937*; *Smith, 1976*; *Barton & Partridge, 2000*; *Lanfear, Kokko & Eyre-Walker, 2014*; *Rousselle et al., 2020*). Heritable variation is

Corresponding author
Mikhail Moldovan,
mika.moldovan@gmail.com

generated mainly by the mutational process (*Lewontin, 1964*; *Avery & Hill, 1977*; *Lynch & Walsh, 1998*). Hence, the mutation rate may be a factor affecting the evolution rate, which we, following J. Maynard Smith, define here as the rate of accumulation of beneficial mutations (*Smith, 1976*; *Nam et al., 2017*; *Rousselle et al., 2020*). As shown recently, in populations with low genetic variability the mutation rate is indeed correlated with the evolution rate (*Rousselle et al., 2020*). Thus, in order to adapt, a low-polymorphic population may need additional expressed genetic variability. Here, we test the hypothesis that a potential source of such variability could be introduced by heritable epigenetic modifications, specifically, mRNA editing (*Bass & Weintraub, 1988*; *Gommans, Mullen & Maas, 2009*; *Klironomos, Berg & Collins, 2013*; *Kronholm & Collins, 2015*).

We consider the A-to-I mRNA editing, where adenine (A) is modified to inosine (I) that is subsequently read by the translation machinery as guanine (G) (*Bass & Weintraub, 1988*). In most of the studied organisms, the A-to-I editing affecting protein sequences is restricted to only a few thousand adenines, with the vast majority of edited adenines located in non-coding regions, for example, in Alu-repeats (*Kim, 2004*; *Ramaswami et al., 2012*; *Yablonovitch et al., 2017*). Edited sites are poorly conserved between species, suggesting that most editing events are non-functional, with a few possible exceptions (*Yang et al., 2008*; *Pinto, Cohen & Levanon, 2014*; *Yu et al., 2016*). However, in coleoids, soft-bodied cephalopods, about 1% of adenines in the transcriptome are edited, and re-coding (i.e., affecting the amino acid sequence) and conserved sites comprise considerable fractions (*Alon et al., 2015*; *Liscovitch-Brauer et al., 2017*). One explanation for this phenomenon comes from the observation that the conserved editing sites tend to be edited in the nervous tissue, and editing may contribute to the increased plasticity and complexity of the coleoid nervous system and behavior compared to other extant cephalopods (*Nautilus*) (*Albertin et al., 2015*; *Alon et al., 2015*; *Liscovitch-Brauer et al., 2017*; *Eisenberg & Levanon, 2018*). This hypothesis is supported by analogous observations in other organisms (*Pinto, Cohen & Levanon, 2014*; *Yu et al., 2016*) and, although indirectly, by the finding that the A-to-I RNA editing has emerged approximately at the same time as the nervous systems of multicellular organisms have become more complex (*Jin, Zhang & Li, 2009*).

A-to-I editing is not absolutely efficient and, if it occurs at a non-synonymous site, would result in two non-identical proteins with a varying ratio (*Gommans, Mullen & Maas, 2009*; *Liscovitch-Brauer et al., 2017*; *Yablonovitch et al., 2017*). The efficiency of mRNA editing depends on the strength of the site motif and the local mRNA secondary structure (*Morse, Aruscavage & Bass, 2002*; *Reenan, 2005*; *Gommans, Mullen & Maas, 2009*; *Alon et al., 2012*; *Savva, Rieder & Reenan, 2012*; *Klironomos, Berg & Collins, 2013*; *Rieder et al., 2013*; *Liscovitch-Brauer et al., 2017*). As the sequence and structure requirements seem to be relatively weak, mRNA editing sites have been proposed to constantly emerge at random points of the genome (*Gommans, Mullen & Maas, 2009*; *Xu & Zhang, 2014*).

To date, four models of A-to-I editing site evolution have been proposed. (i) Most A-to-I editing sites generally are not adaptive and mainly arise at positions with tolerable, that is, effectively neutral or mildly deleterious, A-to-G substitutions (*Xu & Zhang, 2014*).

(ii) A-to-I editing is a mechanism of rescuing deleterious G-to-A substitutions (*Jiang & Zhang, 2019*). (iii) A-to-I editing, generating multiple protein variants, is important for the advantageous transcriptome diversification, and hence the individual sites should be conserved (*Liscovitch-Brauer et al., 2017*; *Eisenberg & Levanon, 2018*). (iv) The potential of A-to-I editing to mimic A-to-G substitutions is advantageous, and thus A-to-I editing sites function as transitory states when an advantageous mutation has not yet occurred (*Popitsch et al., 2020*).

Editing site evolution in *Drosophila* and human has been recently shown to adhere to model (iv) (*Popitsch et al., 2020*), while editing sites in coleoids are largely considered as means for proteome diversification as in model (iii) (*Liscovitch-Brauer et al., 2017*; *Eisenberg & Levanon, 2018*) or be selectively neutral (*Jiang & Zhang, 2019*). We attempt to resolve this controversy by detailed analysis of substitution patterns and selection regimes, taking into account the varying strength of A-to-I editing at different sites.

Generally, in a population with low genetic variability, one might expect evolutionary benefits of A-to-I editing consistent with model (iv). Indeed, if there is a position in the genome occupied by an adenine, but guanine in this position would yield a fitter genotype, there are two evolutionary pathways for adaptation: through an A-to-G substitution at this site, or through emergence of a local sequence context yielding or reinforcing A-to-I editing of this site. If the selective benefit conferred by both pathways is comparable, which of them will be taken will depend on the probability of the corresponding mutation (*Yampolsky & Stoltzfus, 2001*). A specific mutation is needed in the first scenario; by contrast, many different editing context-improving mutations could yield a fitter genotype, and the waiting time for any such mutation could be shorter (*Durrett & Schmidt, 2008*). As a result, selection would lead to emergence of the adaptive editing phenotype.

We propose that non-conserved coleoid A-to-I mRNA editing sites, comprising the larger percentage relative to the conserved ones, could function as substitutes of beneficial A-to-G substitutions in low-polymorphic coleoid populations. We show that the levels of cephalopod A-to-I editing heavily depend on the sequence of adjacent regions, and hence are influenced by a multitude of possible mutations. Critically, we show that edited adenines are more frequently substituted in related species to guanines and less frequently, to cytosines or thymines, than non-edited ones. At strongly edited sites, the adenine-to-guanine transitions are favored by positive selection. Our results suggest that, while conserved coleoid editing sites could be functionally important per se, a large subset of NCES could play a role in the adaptive evolution by introducing, at least in a fraction of transcripts, guanines that are beneficial at the given positions. When this study had been completed, a similar observation was made for *Drosophila* and human editing sites by analysis of genomic polymorphisms (*Popitsch et al., 2020*). This indicates that A-to-I editing could have similar, important evolutionary roles in multiple metazoan lineages.

## MATERIALS AND METHODS

### Data

Transcriptomes for all six considered species, *O. vulgaris*, *O. bimaculoides*, *S. esculenta*, *L. pealei*, *N. pompilius*, and *A. californica*, parameters of editing sites, and tables of

conserved editing sites were taken from the online supplementary data of *Liscovitch-Brauer et al. (2017)* (Fig. 1). Genomic read data were downloaded from the SRA database. *S. esculenta* and *O. vulgaris* genomic read data were taken from bioproject PRJNA299756, *L. pealei*, from PRJNA255916, and *O. bimaculoides*, from PRJNA270931.

## Annotation of structured and unstructured regions

To estimate the structural potential of each position we used *Z*-score values obtained by the RNASurface program (*Soldatov, Vinogradova & Mironov, 2013*). Here, *Z*-score of a sequence is defined as $Z = (E - \mu)/\sigma$ where $E$ is the minimal free energy of a biological sequence, $\mu$ and $\sigma$ are the mean and standard deviation of the energy distribution of shuffled sequences with preserved length and average dinucleotide composition. The program was run with parameters maximal sliding window length 350 and minimal sliding window length 20. From the RNASurface output, structural potential of overlapping segments was inferred. Each position of each transcript was assigned the best (minimal) *Z-score* of all structured segments containing it, if it was less than −2, otherwise it was assigned null value. As a result, each transcript was divided into structured and unstructured regions with a *Z*-score value assigned to all positions in the structured regions. The difference between the structural potential upon the A-to-G change (Fig. 2D) was considered if its absolute value exceeded 2.

## Analysis of polymorphisms

Genomic reads were mapped onto transcriptomes with bowtie2 (*Langmead & Salzberg, 2012*) using the sensitive-local run mode. After the sorting of the resulting read alignment files with the samtools package (*Li et al., 2009*), diploid genotypes were called with bcftools (*Narasimhan et al., 2016*). Next, we discarded all non-SNP variants and variants with the quality score below 20. We computed synonymous nucleotide diversity $\pi_s$ with the pairwise haplotype comparison implemented in the PAML package (*Yang, 2007*).

## Alignments

To construct multiple transcriptome alignments, we selected a transcriptome of one species and performed BLASTn (*Altschul et al., 1990*) with the E-value threshold of $10^{-15}$ against the transcriptomes of the remaining species. Resulting alignment was obtained by merging of the pairwise BLASTn alignments. The results showed only a negligible dependance on the choice of the seed species.

## Context analysis

Site LOGOs were built with the WebLOGO server (*Crooks, 2004*). *R* values for mismatches in contexts of NCES were defined as:

$$R_{N_1,N_2}^{\pm 1} = \frac{p(\mathrm{EN}_1, \mathrm{AN}_2)}{p(\mathrm{AN}_1, \mathrm{AN}_2)}$$

where $N_1$ and $N_2$ represent nucleotides in positions +1 and −1 relative to the considered adenine, $p(\mathrm{EN}_1, \mathrm{AN}_2)$ is the probability of a mismatch at position +1 or −1 relative to the

considered adenine that is edited in one of the two considered species and not edited an another, defined as:

$$p(\mathrm{E}N_1, \ \mathrm{A}N_2) \ = \ \frac{\#(\mathrm{E}N_1, \mathrm{A}N_2)}{\#(E,A)}$$

with $\#(E,A)$ and $\#(\mathrm{E}N_1, \mathrm{A}N_2)$ being the number of homologous A-E states and the number of contextual $N_1$–$N_2$ mismatches associated with the A–E pairs, respectively.

$p(\mathrm{A}N_1, \mathrm{A}N_2)$ is the respective probability when both homologous adenines are non-edited defined as:

$$p(\mathrm{A}N_1, \ \mathrm{A}N_2) \ = \ \frac{\#(\mathrm{A}N_1, \mathrm{A}N_2)}{\#(A,A)}$$

$\#(A,A)$ and $\#(\mathrm{A}N_1, \mathrm{A}N_2)$ being the number of homologous A-A states and the number of $N_1$–$N_2$ mismatches adjacent to the A–A pairs, respectively.

The statistical significance of the $R$ values was assessed by the chi-squared contingency test with the Bonferroni correction on the number of $N_1$–$N_2$ mismatch types. A $2 \times 2$ Contingency matrix $S$ used in the chi-squared test was constructed from the numbers used to define $p(\mathrm{E}N_1, \mathrm{A}N_2)$ and $p(\mathrm{A}N_1, \mathrm{A}N_2)$:

$$S = \begin{pmatrix} \#(\mathrm{E}N_1, \mathrm{A}N_2) & \#(\mathrm{A}N_1, \mathrm{A}N_2) \\ \#(E,A) & \#(A,A) \end{pmatrix}$$

## Substitution matrix

For a considered species, we considered its closest relative and an outgroup that could be either of the two remaining coleoids (Fig. 1A). Given the low number of available species, we used maximum parsimony (MP) to reconstruct ancestral states. Thus, for a position in the alignment, the ancestral state of nucleotide $N$ was inferred if the closest relative and an outgroup had the same nucleotide $N^{\mathrm{anc}}$; an ancestral adenine was considered to be edited if the homologous adenines in the closest relative and an outgroup were edited. The substitution matrix was thus comprised of counts inferred by MP, $\#(N^{\mathrm{anc}} \rightarrow N)$.

## $R$ and $Q$ calculation for non-synonymous and synonymous editing sites

When $R$ measures were computed separately for synonymous (SES) and non-synonymous (NES), we applied a modification of the expression for the $R$ value. For substitutions at SES:

$$R^{\mathrm{syn}}_{\rightarrow N} = \frac{p(\mathrm{E}^{\mathrm{syn}}, \ N)}{p(\mathrm{A}^{\mathrm{syn}}, \ N)}$$

where $\mathrm{E}^{\mathrm{syn}}$ are synonymous editing sites, that is, edited adenines that, when substituted to guanine, do not change the amino acid, and, similarly, $\mathrm{A}^{\mathrm{syn}}$ are synonymous unedited adenines. An analogous formula was applied for non-synonymous editing sites. The definitions of probabilities $p(\mathrm{E}^{\mathrm{syn}}, \ N)$ and $p(\mathrm{A}^{\mathrm{syn}}, \ N)$ are in this case analogous to those used in the context analysis, see above.

When we calculated $R_{N\to}$ separately for NES and SES, we applied another modification of the expression for the $R$ value. For mutations to SES we have:

$$R_{N\to}^{\mathrm{syn}} = \frac{p^*(N^{\mathrm{syn}},\ E^{\mathrm{syn}})}{p^*(N^{\mathrm{syn}},\ A^{\mathrm{syn}})}$$

where $E^{\mathrm{syn}}$ and $A^{\mathrm{syn}}$ are defined just as above, and $N^{\mathrm{syn}}$ represents nucleotides, that, when substituted with adenine and with guanine would yield the same amino acid. $p^*$ are, just like in formula (3), conditional probabilities:

$$p^*(N,\ E) = p(N,\ E) \Big/ \frac{\#E}{\#E + \#A}$$

$$p^*(N,\ A) = p(N, A) \Big/ \frac{\#A}{\#E + \#A}$$

An analogous formula is applied to NES, with $N^{\mathrm{non}}$ representing nucleotides, that, when substituted with adenine and with guanine would yield different amino acids.

## Calculation of dN/dS

We estimated the strength of positive selection acting on substitutions to guanines and to pyrimidines separately by applying the *dN/dS* measure to edited adenines with a subsequent normalization by *dN/dS* of unedited adenines. Thus, for substitutions to G we applied the formula (Fig. S9):

$$\frac{dN(E \to G)}{dS(E \to G)} \Big/ \frac{dN(A \to G)}{dS(A \to G)}$$

where *dN* were calculated for all codons and *dS*, for four- and six-fold degenerate codons. An analogous formula was used to estimate selection acting on E-to-Y substitutions. Next, we applied this measure separately for 10% editing level (EL) bins, counted Pearson's correlation coefficient and applied the *F* statistic to estimate the significance of the obtained correlation.

Positive selection on editing sites was estimated with the *dN/dS* ratio where non-synonymous substitutions were considered for edited adenines, and synonymous, for unedited adenines:

$$\frac{dN(E \to G)}{dS(A \to G)} = \frac{p(E^{\mathrm{non}} \to G)}{p(A^{\mathrm{syn}} \to G)} * \frac{\xi^{\mathrm{non}}}{\xi^{\mathrm{syn}}}$$

where $\xi^{\mathrm{non}}$ and $\xi^{\mathrm{syn}}$ are normalizing coefficients accounting for differences in codon probabilities and different probabilities of, respectively, synonymous and non-synonymous substitutions under the neutral evolution assumption. These coefficients are defined as:

$$\xi^{\mathrm{non}} = 1 \Big/ \left( \sum_{N_1 N_2 N_3\ \in\ \{A,T,G,C\}^3} f(N_1 N_2 N_3) \times K^{\mathrm{non}}(N_1 N_2 N_3,\ A \to G) \right)$$

$$\xi^{\text{syn}} = 1 \bigg/ \left( \sum_{N_1 N_2 N_3 \,\in\, \{A,T,G,C\}^3} f(N_1 N_2 N_3) \times K^{\text{syn}}(N_1 N_2 N_3, \ A \rightarrow G) \right)$$

where $f(N_1 N_2 N_3)$ is the codon frequency while $K^{\text{non}}$ and $K^{\text{syn}}$ are, respectively, the numbers of possible non-synonymous and synonymous A-to-G substitutions in a given codon.

## Statistics

For mutation frequency and *dN/dS* analysis, statistics were obtained from $10^5$ random sets of mutation numbers sampled from the binomial distributions with the parameters equal to the observed substitution frequencies. For the analysis of parallel evolution, two-tailed confidence intervals were inferred from the binomial distribution. The binomial test was applied to compare fractions of conserved and not conserved editing sites in structured segments. For the analysis of changes in the secondary structure stability following A-to-G in silico substitutions, sizes of tails in the distribution of *Z*-score differences were compared using the binomial test, and to compare the results for different types of sites, random 100-sequence samples of each type were compared with the Wilcoxon signed-rank test.

## Data availability

Ad hoc scripts were written in Python. Graphs were built using R. All scripts and data analysis protocols are available online at https://github.com/mikemoldovan/coleoidRNAediting.

## RESULTS

### Editing level is associated with the local and global sequence context

We studied the A-to-I editing using available genomic read libraries, transcriptomes, and editing sites data for four coleoids, closely related octopuses *Octopus vulgaris* and *O. bimaculoides*, squid *Loligo pealei*, and cuttlefish *Sepia esculenta* (*Liscovitch-Brauer et al., 2017*). As outgroups, we considered nautiloid *Nautilus pompilius* and gastropod mollusk *Aplysia californica* (*Liscovitch-Brauer et al., 2017*) (Fig. 1A).

The action of editing sites as surrogates of beneficial A-to-G substitutions presumes advantageous enhancement of editing probabilities at individual sites. As A-to-I editing is affected by the local sequence context (*Alon et al., 2012*; *Liscovitch-Brauer et al., 2017*) and the RNA secondary RNA structure (*Morse, Aruscavage & Bass, 2002*; *Reenan, 2005*; *Gommans, Mullen & Maas, 2009*; *Savva, Rieder & Reenan, 2012*; *Klironomos, Berg & Collins, 2013*; *Rieder et al., 2013*), one would expect, firstly, contextual differences around weakly vs heavily edited sites and, secondly, contextual mutations associated with changes in editing status. Indeed, we have observed a previously unnoted dependance of the editing level (EL) (Fig. 1B), defined as the percent of transcripts containing I at the considered site at the moment of sequencing (Fig. 2A; Fig. S1), on the site context (±1 motif). Certain changes in the ±1 motif, specifically, an increase in the preference for G

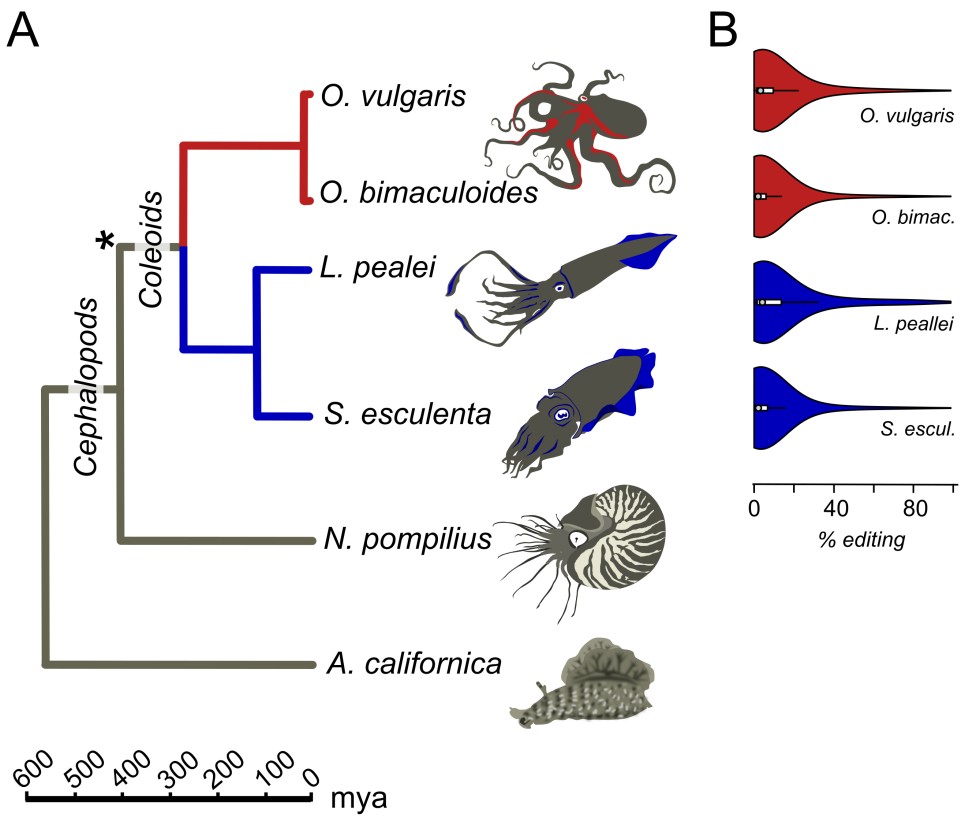

**Figure 1 Prevalent mRNA editing in coleoid mollusks.** (A) Phylogenetic tree of the species taken from TimeTree (*Hedges, Dudley & Kumar, 2006*). The asterisk marks the putative beginning of editing site expansion (*Liscovitch-Brauer et al., 2017*). (B) Distributions of per-nucleotide editing levels of the predicted editing sites in the studied coleoids.

or T at the +1 position, are associated with the increase of EL, although its information content of the motif remains approximately the same. Although the ±1 motif of both weakly and strongly edited sites is consistent with the ADAR (adenosine deaminases acting on RNA) profile (*Alon et al., 2012*; *Liscovitch-Brauer et al., 2017*), this observation could point to the action of different ADAR enzymes or to different modes of action of the same enzyme on strongly and weakly edited sites. There also seem to be some differences between the motifs of conserved and non-conserved sites (Fig. S2).

The analysis of NCES in the octopus pair demonstrates overrepresentation of mismatches in the ±1 motif of the edited adenines reinforcing the local editing context compared to the homologous unedited adenines, for which the editing context is not observed (See "Materials and Methods", Fig. S3). Thus, both the editing status and the EL of a site are associated with substitutions in the ±1 motif. In the squid-cuttlefish pair, the higher number of mutations obscures this analysis.

To estimate the size of the region that affects editing, we have measured the correlation between the editing level difference in conserved editing sites (CES) in closely related species and the number of mismatches in variable-sized windows centered at edited adenines. The window size yielding the largest correlation coefficient shows the average span

of the context affecting the ADAR performance. For the *Octopus* pair, the highest correlation has been obtained at the window size of ~100 nucleotides (Fig. S4), consistent with previous estimates for the length of the region affecting editing (*Liscovitch-Brauer et al., 2017*).

## Editing level is affected by secondary structure in adjacent RNA

The A-to-I editing in model species depends on large RNA structures spanning hundreds of nucleotides in addition to the local sequence context (*Morse, Aruscavage & Bass, 2002*; *Reenan, 2005*; *Enderö et al., 2009*; *Rieder et al., 2013*; *Kurmangaliyev, Ali & Nuzhdin, 2015*) as the ADAR-mediated mRNA editing generally requires secondary RNA structures (*Gommans, Mullen & Maas, 2009*; *Farajollahi & Maas, 2010*; *Xu & Zhang, 2014*).
Thus, we have assessed the link between RNA secondary structure and ELs of focal sites.

We have predicted structured segments in the transcripts of all six considered species. As the fraction of adenines located within structured segments is the same for all cephalopod species, including *Nautilus* (Fig. S5), our secondary structure analyses are not systematically influenced by the GC-content of the studied genomes (*Wang et al., 1984*). Then we have assessed the contribution of mRNA secondary structure to the editing process by comparing structural contexts of edited and unedited adenines (Materials and Methods). The fraction of edited adenines located in putative structured regions is higher than the respective fraction for non-edited sites. Moreover, sites that are more highly edited (Fig. 2B) as well as sites conserved between more distant species (Fig. 2D) tend to be more structured.

To uncover the connection between the strength of a local secondary RNA structure and the editing status at individual sites, we have compared the fractions of NCES located within structured segments in edited *vs.* non-edited states. We considered the *Octopus* pair and the squid–cuttlefish pair. For both pairs, we have compared CES and NCES.
For NCES in both species pairs we have observed significantly more cases when the edited site in a pair is more structured than the unedited one while the control CES set shows no bias (binomial test $p < 10^{-3}$ for all pairs; Fig. 2C; Fig. S6).

Not only the fact of editing, but the difference in editing levels is linked to local secondary structures. For the closely related *Octopus* pair, we have calculated correlations between differences in ELs of homologous edited adenines and differences in their structure $Z$-scores (Fig. S7). Almost no correlation ($r = 0.1$, $t$-test $p < 0.05$) is seen when the EL difference is small (>5%), whereas for large differences in ELs (>50%) the correlation is substantial ($r = 0.7$, $t$-test $p < 0.05$). A likely explanation is that larger differences are indeed due to the strength of the local secondary structure, whereas small differences in ELs arise as consequences of random noise. Consistent with the observations above, if we consider structures around edited adenines and their unedited homologs, setting the ELs of unedited adenines to 0, we observe a similar, although a weaker trend (Fig. S7), with correlations reaching 0.4 ($t$-test $p < 0.05$) when the ELs of NCES are high.

The observations about local contexts, both the ±1 motif and RNA structures, imply that mutations near editing sites influence the editing status as well as the editing level.

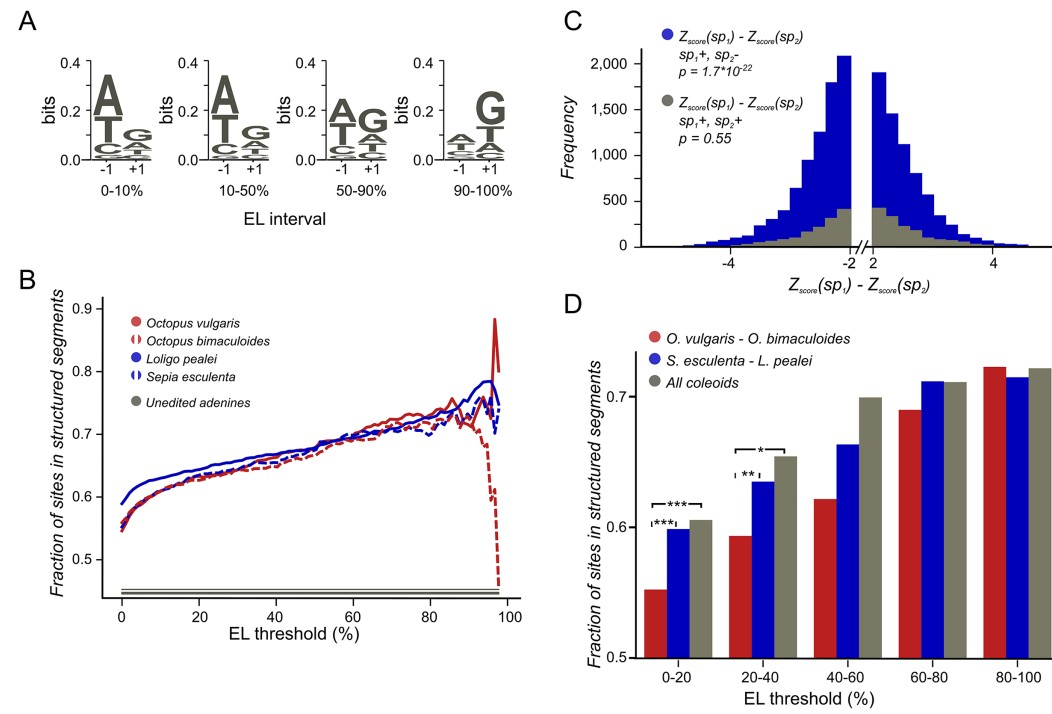

**Figure 2 Coleoid editing site contexts.** (A) *O. bimaculoides* editing site context changes with the increase of editing level. The height of the letters represents the LOGO bit score of each nucleotide. (B) Highly conserved editing sites tend to be relatively more structured. The fraction of editing sites that are in structured segments is shown for different editing levels: red—*O. vulgaris*, red dashed—*O. bimaculoides*, blue—*L. pealei*, blue dashed—*S. esculenta*, gray—the constant showing the fraction of unedited adenines located in structured segments. The noisy pattern at the right is due to a low number of very highly edited sites. (C) The stability of the local secondary structure is higher at edited adenines than at homologous, non-edited adenines for the squid/cuttlefish pair. The distribution of the difference of the minimal free energy *Z*-score between homologous sites in squid and cuttlefish is shown in blue when two homologous sites have different editing status (edited minus unedited) and in gray when both sites in a pair are edited. The left tail of the blue histogram is heavier than the right one ($p = 9.37 \times 10^{-33}$ vs 0.32 for the gray histogram), showing that the editing sites tend to regions with higher secondary structure stability. (D) Conserved editing sites tend to be more structured. The three groups of sites are those present in two of the four species (*O. vulgaris* and *O. bimaculoides*, red, or *L. pealei* and *S. esculenta*, blue), or in all four species (grey). Statistically significant differences are shown with brackets (***$p < 0.001$, *$p < 0.05$).   

## Edited adenines are often substituted by guanines

If edited adenines indeed frequently mimic the beneficial guanine state, the substitution patterns of edited and unedited adenines should differ, with edited adenines being more prone to substitutions to guanine and less prone to substitutions to cytosine or thymine (*Popitsch et al., 2020*). Firstly, we performed the analysis of the species pairs to infer the properties of A-G mismatches at editing sites. For a pair of considered species, we define $R$ as the mismatch probability for an edited adenine divided by the probability of the same mismatch for an unedited adenine: $R_N = p(E,N)/p(A,N)$, where E and A are, respectively, edited and not edited adenines in one species, and $N$ is the non-E, non-A nucleotide at the homologous site in the other organism. Similar formulas are applied when we consider directed substitutions instead of mismatches. If a pair of the ancestral

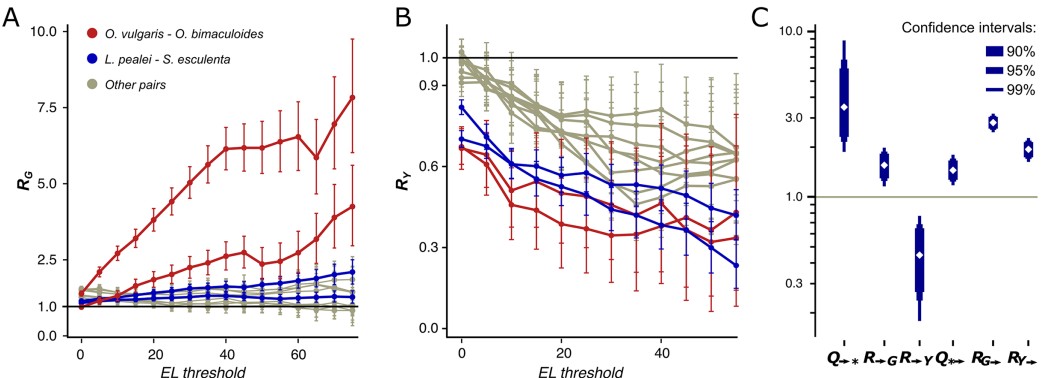

**Figure 3** *R* and *Q* values. Dependence of $R_G$ (A) and $R_Y$ (B) on the editing level. Two curves for each pair are given, since $R_N$ is calculated two times for each pair of species using one of them a a reference each time. The red curves correspond to the pair *O. vulgaris–O. bimaculoides*; the blue curves, to the pair cuttlefish–squid, the gray curves, to distant pairs. (C) Mutational characteristics of editing sites for the squid–cuttlefish summary substitution matrix. Left to right: $Q_{\rightarrow*}\gg1$, $R_{\rightarrow G}>1$, $R_{\rightarrow Y}<1$, $Q_{*\rightarrow}>1$, $R_{G\rightarrow}\gg1$, $R_{Y\rightarrow}>1$.

and the descendant species is considered, we use notation $R_{\rightarrow N}$ to identify the directionality. $R_{\rightarrow N} = p(E{\rightarrow}N)/p(A{\rightarrow}N)$, where $p(E{\rightarrow}N)$ and $p(A{\rightarrow}N)$ are, respectively, the probabilities of the substitution of the edited and non-edited adenine to N. Similarly, notation $R_{N\rightarrow}$ is used when substitutions from ancestral N to E and A are considered: $R_{N\rightarrow} = p(N{\rightarrow}E)/p(N{\rightarrow}A)$. Higher values of $R_{N\rightarrow}$ imply that the ancestral nucleotide N is more likely to be substituted by an edited adenine, compared to an unedited one.

We have observed a striking dependance of the calculated mismatch probabilities on the editing status of the adenines and their ELs. In the *Octopus* pair, $R_G$ and $R_Y$ (Y denotes pyrimidine, C or T) differ both in value and in the dependance on the EL (Figs. 3A and 3B). Indeed, $R_G$ is always higher than $R_Y$ with $R_G$ further increasing and $R_Y$ decreasing as the EL increases. The probability for an adenine to be substituted by a guanine in the *O. vulgaris* lineage is ~8 times higher when the homologous adenine is strongly edited in *O. bimaculoides* than when it is not (Fig. 3A). For the more distantly related squid–cuttlefish pair, we observe a similar although less pronounced effect. For all distant pairs, that is, *Octopus*–squid/cuttlefish, $R_G$ shows no or only a weak dependance on the EL.

We have calculated *R* values separately for NES, which comprise between 64.6% and 65.7% of all detected coleoid editing sites, and for SESs which comprise the remaining 34.3–35.4%. NES (Figs. S8A and S8B) demonstrate the same pattern as described above for all sites, whereas for SES, we see no dependance of $R_Y$ on the EL (Figs. S8C and S8D). NES demonstrate very low $R_Y$ at high ELs. These patterns suggest that at highly edited nonsynonymous adenine sites, any nucleotide other than guanines are impeded by strong negative selection; whereas the guanine states at such sites are frequent. Thus, at non-synonymous NCES, the selection patterns differ from those at non-conserved adenines: Y mismatches with NCES experience stronger negative selection than Y mismatches with non-edited adenines, and stronger positive and/or weaker negative selection acting on E-to-G or G-to-E substitutions compared to A-to-G or G-to-A ones, respectively.

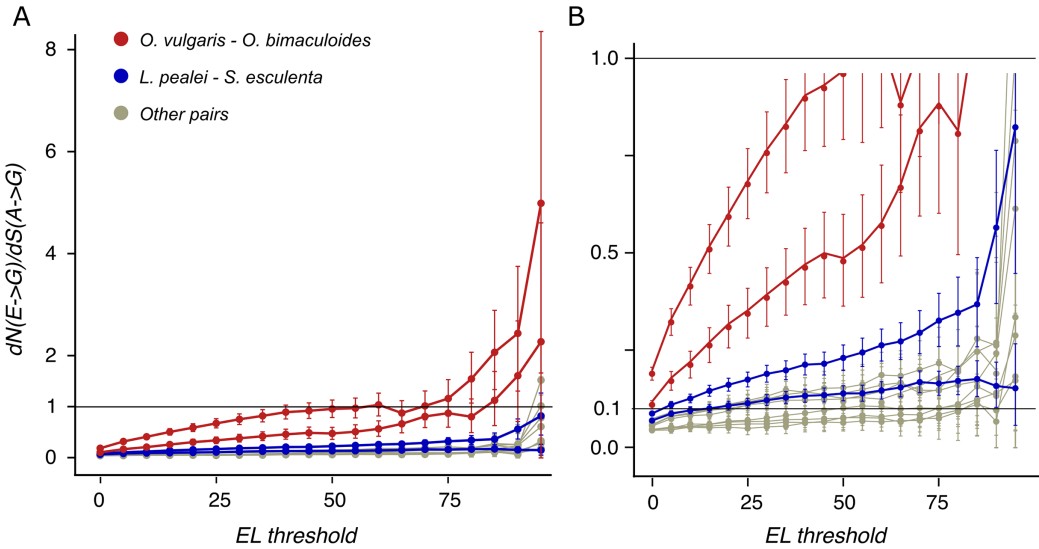

**Figure 4** *dN/dS* **values of adenine substitutions to guanines for various EL thresholds.** Non-synonymous substitutions are calculated for edited adenines, and synonymous, for unedited adenines. Error bars indicate the 95% probability value range. (A) Plot for the whole range of *dN/dS* values. (B) Truncated value range. Note the increase of *dN/dS* values at high EL values for all species pairs.

## Editing recapitulates substitutions that are positively selected

To reveal the mode of selection at edited sites, we have calculated the *dN/dS* ratios separately for mismatches of edited and unedited adenines with guanines and with pyrimidines (Fig. S9). For weakly edited adenines, the *dN/dS* values of mismatches with guanines and with pyrimidines are approximately the same as those for unedited adenines. However, at highly edited sites, the *dN/dS* ratio for substitutions to guanine is two- to threefold higher, compared to unedited adenines, while the respective ratio for pyrimidines is twofold lower. Thus, strongly edited sites evolve under weaker purifying selection against E-to-G and/or G-to-E transitions and stronger purifying selection against E-to-Y and/or Y-to-E substitutions.

To distinguish between positive selection and relaxation of negative selection at these sites, we have calculated *dN/dS* for A-G mismatches where *dN* and *dS* are calculated for edited and unedited adenines, respectively. It is larger than 1 at high ELs for the closely related octopus species pair (Fig. 4), indicating positive selection acting on the E-to-G transition: heavily edited adenines are positively selected for substitutions to guanine.

## E-to-G substitutions vs G-to-E substitutions

In theory, two processes could lead to the increase in the observed *R* and *dN/dS* values of edited sites—the increased frequency of either E-to-G or G-to-E substitutions.

To distinguish between these possibilities, we use the procedure described in "Materials and Methods" to calculate the frequencies of all types of substitutions for each species since its closest ancestor. We also consider the more robust, averaged substitution frequencies for the *Octopus* pair and for the squid–cuttlefish pair. As the frequencies of substitutions to edited and non-edited adenines are calculated separately, we introduce the normalized,

directional measure $Q_{\to *}$ reflecting the preference of edited adenines to substitute to guanine:

$$Q_{\to *} = \frac{R_{\to G}}{R_{\to Y}} = \frac{p(E \to G)}{p(A \to G)} \bigg/ \frac{p(E \to y)}{p(A \to Y)} = \frac{p(E \to G)}{p(E \to Y)} \bigg/ \frac{p(A \to G)}{p(A \to Y)}$$

By this definition, the $Q_{\to *}$ measure is an indicator of the joint effect of the prevalence of E-to-G over A-to-G substitutions and of the underrepresentation of E-to-Y relative to A-to-Y substitutions. For the squid–cuttlefish clade, and for SESs and NESs considered separately, $Q_{\to *}$ ranges from 3.49 to 6.4 (Fig. 3C; Figs. S10A and 10B), in all cases being significantly higher than 1 expected under a neutral model ($p < 0.005$). Hence, as in the case of pairwise comparison of extant species (Figs. 3A and 3B), edited adenines have a substitution pattern strikingly different from that of unedited adenines, and are likely to mutate into guanines.

However, large values of $Q_{\to *}$ may be explained by two effects, high $R_{\to G}$ of E-to-G substitutions or low $R_{\to Y}$ of E-to-Y substitutions (Fig. 3C) both yielding $R_{\to G}$ higher than $R_{\to Y}$. $R_{\to G}$ is higher than 1 ($p < 0.005$), thus indicating that an edited adenine is more likely to be substituted by guanine than an unedited adenine. Combined with $R_{\to Y}$ being smaller than 1 ($p < 0.005$), this indicates that in fact both effects contribute to the observed $Q_{\to *}$ values. A similar pattern holds if we consider NES and SES separately: $R_{\to G}$ is higher than $R_{\to Y}$, although for NES high $Q_{\to *}$ can be almost entirely attributed to $R_{\to G}$, and for SES, to $R_{\to Y}$ ($p < 0.005$) (Figs. S10A and 10B).

To analyze the directionality of the mutation process that affects editing states, we consider a similar function measuring the degree of prevalence of G-to-E substitutions:

$$Q_{* \to} = \frac{R_{G \to}}{R_{Y \to}} = \frac{p * (G \to E)}{p * (G \to A)} \bigg/ \frac{p * (Y \to E)}{p * (Y \to A)} = \frac{p * (G \to E)}{Y * (G \to E)} \bigg/ \frac{p * (G \to A)}{p * (Y \to A)}$$

where probabilities $p^*$ are conditional probabilities of a nucleotide mutating to either edited or unedited adenine after taking into account differences in the E and A densities in the transcriptomes:

$$p^*(N \to E) = p(N \to E) \bigg/ \frac{\#E}{\#E + \#A}$$

$$p^*(N \to A) = p(N \to A) \bigg/ \frac{\#A}{\#E + \#A}$$

For both the *Octopus* pair and the squid–cuttlefish pair, $Q_{* \to}$ is larger than 1 ($p < 0.005$) (Fig. 3C; Fig. S10), thus suggesting that guanines tend to be substituted by edited rather than unedited adenines. However, this effect is on average twofold smaller than that for substitutions of edited adenines to guanines, suggesting that the process of G-to-E transitions is less directional than that for E-to-G transitions. If $R_{G \to}$ and $R_{Y \to}$ are considered separately, they both are larger than the expected value 1 (Fig. 3C) ($p < 0.005$), which points to a generally faster generation of E sites from both G and Y nucleotides. As the observed effect is small, it could be attributed to weaker negative selection acting

upon the G-to-E transition relative to G-to-A, as the edited adenine is a state closer to the guanine-only variant (*Jiang & Zhang, 2019*).

The $Q$ value defined as the ratio of undirected $R$ values increases with the EL (as follows from Figs. 3A and 3B). On the other hand, formally it is monotonic with respect to the directed $Q_{\to*}$ and $Q_{*\to}$ values (see Supplmental Material 1). Hence, even though we could not detect a significant dependance of $Q_{\to*}$ and $Q_{*\to}$ on EL due to insufficient data, at least one of them should increase with the EL. However, the effects observed for the E-to-G substitution are more pronounced compared with those for the G-to-E substitutions, hinting at A-to-I editing sites mimicking beneficial A-to-G substitutions rather than rescuing deleterious G-to-A substitutions.

## DISCUSSION

### The hypothesis about the adaptivity of non-conserved editing sites is supported by our observations

Editing in coleoids is essential for transcriptome diversification, and results in a more complex phenotype (*Liscovitch-Brauer et al., 2017*; *Eisenberg & Levanon, 2018*). Indeed, a considerable fraction of coleoid editing sites are conserved between even distantly related species, and a majority of heavily edited sites affect protein sequence (*Liscovitch-Brauer et al., 2017*). We propose that non-conserved coleoid editing sites could facilitate adaptation by extending selection to regions affecting editing if guanine is the beneficial variant at the editing site. This hypothesis is directly supported by our observations. Indeed, strong dependance of editing on the local context allows for selection of mutations in the vicinity of the editing site, hence extending the variety of beneficial mutations. On the other hand, edited adenines indeed tend to be substituted by guanines, and guanines are selected for if the editing levels of homologous adenines is high. This positive selection pattern is specific to guanine variants, as substitutions of edited adenines to cytosine or thymine are avoided.

An indirect observation also supports our hypothesis. Sizes of the effects such as the E-to-G substitution rate or the rate of positive selection on the guanine variant at editing sites are larger for heavily edited adenines compared to medium and weakly edited ones. This effect could be explained by beneficial A-to-G substitutions provoking selection on adjacent regions, which leads to the increased ELs and hence to the enhanced presence of the guanine-like variant. Indeed, if G is beneficial at a given site, it would manifest as both positive selection towards G at this site, and by mutations at adjacent sites yielding higher A-to-I editing level, and hence these two types of effects would be correlated.

### Positive selection in favor of E-to-G substitutions

Why would substitutions that recapitulate editing be adaptive? Conceivably, it could be that variability at the transcriptome level is advantageous by itself, contributing to the proteome diversity, similar to alternative splicing, alternative transcription and translation starts, etc (*Raj & Van Oudenaarden, 2008*; *Gommans, Mullen & Maas, 2009*; *Pickrell et al., 2010*). However, this scenario does not explain positive selection of substitutions to G.

Alternatively, editing might create an unconditionally beneficial variant, so that at an edited site, G is always better than A. Under this scenario, editing could recreate the G allele previously lost due to a deleterious G-to-A mutation, or produce a novel G variant which is favored by selection but has not been present at this site previously (*Jiang & Zhang, 2019*). This scenario is supported by the observed selection favoring guanines at edited sites.

But why would selection in favor of G result in an increased A-to-I editing of a fraction of the transcripts, when a "direct" A-to-G genomic mutation at this site would lead to the same result in 100% of transcripts? One reason could be that mutations creating editing sites and/or increasing editing level are more numerous, and therefore more readily available. For a strongly advantageous mutation (with $4N_e s \gg 1$) that does not preexist in the population, the time till its fixation equals $1/(4N_e s\mu)$, where $N_e$ is the effective population size, $s$ is selection in favor of the new mutation, and $\mu$ is the mutation rate, see eq. 3.22 in *Kimura, 1983*. If two types of mutations can yield the desired phenotype, which of them would be the first to fix in an evolving lineage is determined by the product of the corresponding selection and mutation rates.

Let $\mu_1$ be the rate of the direct mutation, and $s_1$, selection in its favor. Assume that an increase in the number of favored transcripts can also be achieved by $M$ other mutations, each characterized by rate $\mu_2$ and selection $s_2$. The probability that the editing-enhancing mutation will be the first to occur then equals $M\mu_2 s_2/(\mu_1 s_1 + M\mu_2 s_2)$ (*Yampolsky & Stoltzfus, 2001*). If $M\mu_2 s_2 > \mu_1 s_1$, the editing-increasing mutation will typically fix earlier than the direct mutation. As we show, many tens of sites may affect editing, making $M$ large, and this scenario likely. For example, if the direct A-to-G substitution confers a 10% increase in fitness, but a 1% increase can be achieved by changes in editing by mutations at each of 20 other sites, then the editing-increasing change will be the first to occur with probability 2/3 if the mutation rates are uniform.

This reasoning only applies if the within-species variability level $N_e\mu$ is low ($\ll 1$); otherwise each site will carry a preexisting mutation, and the mutation rate will be less relevant (*McCandlish & Stoltzfus, 2014*). Low variability is indeed a characteristic trait of the considered coleoid species, with synonymous-site pairwise divergence of $2.5 \times 10^{-3}$ for *O. bimaculoides*, $2.2 \times 10^{-3}$ for *O. vulgaris*, $1.8 \times 10^{-3}$ for *S. eculenta*, and $4.5 \times 10^{-3}$ for *L. pealei* (see "Materials and Methods"). These values imply $N_e\mu \ll 1$, suggesting that evolution can be indeed mutation-limited in this group of species. Low values of $N_e\mu$ characteristic of higher animals have been proposed to underlie many aspects of genomic complexity (*Lynch, 2007*); they may also cause the high prevalence of RNA editing in coleoids.

When this study had been completed, *Popitsch et al. (2020)* published a population-genetic study of *Drosophila* and human A-to-I RNA editing sites, in which they showed a similar pattern of selection at editing sites, with the derived G state selected upon, whereas C and T variants being suppressed, indicating enhanced negative selection. That study indirectly supports our claim about coleoid A-to-I editing sites mimicking beneficial A-to-G substitutions. Furthermore, as coleoids possess many more conserved re-coding A-to-I editing sites than any other studied metazoan group, one might expect the bulk of

coleoid editing, especially at heavily edited sites, to be important per se, for example, for transcriptome diversification, which would result in suppression of any non-adenine variants in editing sites. On the contrary, we have observed positive selection in A-G mismatches, when adenines are heavily edited, with selection acting specifically on A-to-G transitions. Also, like *Popitsch et al. (2020)*, we have observed enhanced negative selection against A-to-C and A-to-T substitutions and mismatches at coleoid editing sites. The consistency of results obtained for coleoids, *Drosophila*, and human points towards a general role of A-to-I editing sites as imitations and precursors of A-to-G transitions in the evolution of metazoans with low-polymorphic populations.

## Conservation and function of editing

Earlier, it has been proposed that most editing sites result from tolerable promiscuous ADAR action (*Xu & Zhang, 2014*). However, the A-to-I editing sites in coleoids are under positive selection if ELs are high (Fig. 4). Hence large ELs cannot result simply from the tolerance towards substitutions to guanines at these sites.

Coleoid editing sites are often considered to be important for complex regulation (*Albertin et al., 2015*; *Alon et al., 2015*; *Liscovitch-Brauer et al., 2017*; *Eisenberg & Levanon, 2018*; *Jiang & Zhang, 2019*). However, this applies only to conserved, and hence functional, editing sites. We propose that coleoid editing sites form two populations with different properties. Firstly, there are functional editing sites, which are important per se due to their ability to diversify protein products in various tissues and environmental conditions (*Savva, Rieder & Reenan, 2012*; *Alon et al., 2015*; *Harjanto et al., 2016*; *Buchumenski et al., 2017*; *Duan et al., 2017*; *Liscovitch-Brauer et al., 2017*; *Tan et al., 2017*). As such sites should be retained over long periods of time, we may consider conservation as a good proxy for functionality. Conserved sites are surrounded by conserved regions (*Liscovitch-Brauer et al., 2017*), their ELs show dependance on the number of substitutions in adjacent regions (Fig. S4), and they comprise up to about a half of A-to-I editing sites in a coleoid transcriptome (*Liscovitch-Brauer et al., 2017*).

Secondly, there are non-functional sites; the proxy here are non-conserved sites, with a caveat that some recently emerged sites could be functional. Nonetheless, as the proportion of young functional sites should be minimal (*Gommans, Mullen & Maas, 2009*), the general properties of non-conserved sites should reasonably well represent those of non-functional ones. Non-conserved sites are not flanked by conserved regions, their ELs show no correlation with the number of substitutions in adjacent regions, and their sequence contexts differ from those of the conserved ones (Fig. S2). Our hypothesis that (non-conserved) editing sites have an intrinsic evolutionary value does not contradict the fact that some (possibly large) subset of editing sites are functional as editing sites per se from the physiological point of view.

Theoretically, our results could have been influenced by underprediction of editing sites. As the mean EL is about 5%, a site might be easily missed especially in transcripts with low expression levels (*Bahn et al., 2011*; *Alon et al., 2012*; *Liscovitch-Brauer et al., 2017*). However, the majority of our observations are obtained for highly edited adenines, which

are predicted with greater accuracy (*Bahn et al., 2011*), and hence should not be influenced by missing weakly edited sites.

## Theoretical frameworks and alternative explanations

Our results could be interpreted within several paradigms. Firstly, as noted above, the observations could mean that editing rescues deleterious G-to-A substitutions (*Jiang & Zhang, 2019*). However, as also mentioned above, the estimates of $Q$ values, which represent the mutation process directionality, indicate that E-to-G substitutions differ in terms of the transition/transversion rate from A-to-G ones to a much greater extent, than G-to-E substitutions differ from G-to-A ($Q_{\ni*} >> Q_{*\ni}$); in addition, $Q_{*\ni}<1$ at non-synonymous sites (Fig. 3C), again supporting the idea that the E-to-G transitions contribute to the observed effects to a larger degree. Ultimately, this issue would be resolved when more data are available, allowing for the reconstruction of ancestral states of NCES.

Our results could be formulated in terms of Waddington's Genetic Assimilation (*Waddington, 1953a*, *1953b*; *Lynch & Walsh, 1998*; *Crispo, 2007*; *Ghalambor et al., 2007*; *Ghalambor et al., 2015*; *Levis & Pfennig, 2016*; *Ho & Zhang, 2018*; *Levis & Pfennig, 2019*). Editing could buffer coleoids against environmental changes — under novel conditions the phenotype changes (adenine is edited and read as guanine), and subsequently this change is reinforced on the genome level by the selection process, which we observe as positive selection pressure on E-to-G transitions. However, at present we have no data on environmental variance in the coleoid A-to-I editing.

The preadaptation paradigm refers to a pre-existing structure that has changed its function or acquired a new one in the course of evolution (*Darwin, 1872*; *Gould & Vrba, 1982*; *McLennan, 2008*; *Ardila, 2016*; *Casinos, 2017*; *Cadotte et al., 2018*). Here, as non-functional editing should be mostly effectively neutral (*Gommans, Mullen & Maas, 2009*), it might generate a pool of variants, some of which may become advantageous in the future, when the genetic background or environmental conditions change. However, to claim preadaptation one should determine the function of each editing site, which is not feasible.

Hence, the most reasonable framework for our findings seems to be in terms of non-functional editing sites enhancing the expressed genetic variability, thus contributing to the acceleration of the evolutionary process at sites with beneficial A-to-G substitution. The Continuous Probing Hypothesis (*Gommans, Mullen & Maas, 2009*) states that editing sites, due to the lack of a strict context, constantly emerge at random points of the transcribed genomic regions. Hence, an adenine with a beneficial substitution to guanine could become edited if the editing context emerges around it purely by chance. The context can be further selected upon, resulting in the mimicking of the beneficial guanine variant. (An extended version of this discussion is provided as Supplemental Material 3)

A similar rhetoric can be applied to other cellular information transmission processes such as transcription and splicing. These processes depend on regulatory sites and contexts that change the quantity, dynamics (developmental stage, tissue-specificity, response to external conditions), and sequence of encoded proteins and hence are subject to selection

(*Raj & Van Oudenaarden, 2008*; *Pickrell et al., 2010*). Hence a natural extension of this study would be to systematically assess the evolutionary advantage of noise in information transmission processes in low-polymorphic populations.

## CONCLUSIONS

RNA editing sites are much more numerous in soft-bodied cephalopods (coleoids) than in any other studied group (*Liscovitch-Brauer et al., 2017*). Here, we show the capacity of numerous coleoid RNA editing sites to function as surrogates for beneficial A-to-G substitutions. This effect is more pronounced for heavily edited sites. At that, the latter are surrounded by stronger local RNA secondary structure (expectedly) and feature different sequence context (unexpectedly). The RNA structure is even stronger around edited adenines homologous to guanines in sister species. Edited adenines tend to be substituted to guanines, and this tendency is supported by positive selection at highly edited sites.

These observations may be explained by the beneficial effect of increased phenotypic diversity in a low-polymorphic population, enhancing adaptation and facilitating the evolutionary process. Besides, A-to-I editing at sites where G would be preferred provides a larger (than a single nucleotide position) target for mutations increasing the editing level. Together with similar recent observations on *Drosophila* and human editing sites (*Popitsch et al., 2020*), this points at a general role of RNA editing in the molecular evolution of metazoans.

## ACKNOWLEDGEMENTS

The authors are grateful to Stepan Denisov, Sofya Garushyants, Fyodor Kondrashov, and Eugene Koonin for discussions, criticisms, and suggestions.

### Funding

This study was supported by the Russian Foundation of Basic Research under grant 18-29-13011. The funders had no role in study design, data collection and analysis, decision to publish, or preparation of the manuscript.

### Grant Disclosures

The following grant information was disclosed by the authors:
Russian Foundation of Basic Research: 18-29-13011.

### Competing Interests

Prof. Mikhail S. Gelfand is an Academic Editor for PeerJ.

### Author Contributions

- Mikhail Moldovan conceived and designed the experiments, performed the experiments, analyzed the data, prepared figures and/or tables, authored or reviewed drafts of the paper, and approved the final draft.

- Zoe Chervontseva performed the experiments, analyzed the data, prepared figures and/or tables, authored or reviewed drafts of the paper, and approved the final draft.
- Georgii Bazykin conceived and designed the experiments, authored or reviewed drafts of the paper, and approved the final draft.
- Mikhail S. Gelfand conceived and designed the experiments, authored or reviewed drafts of the paper, and approved the final draft.

## Data Availability

All scripts and data analysis protocols employed to obtain results outlined in the article are available at GitHub: https://github.com/mikemoldovan/coleoidRNAediting.

The data employed in this study are available in the Supplemental Materials of *Liscovitch-Brauer et al. (2017)* (Liscovitch-Brauer N, Alon S, Porath HT, Elsteom B, Unger R, Ziv T, Admon A, Levanon EY, Rosenthal JJC, Eisenberg E. Trade-off between Transcriptome Plasticity and Genome Evolution in Cephalopods. Cell. 2017; 169, 191–202.e11; DOI 10.1016/j.cell.2017.03.025).

## Supplemental Information

Supplemental information for this article can be found online at http://dx.doi.org/10.7717/peerj.10456#supplemental-information.

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
