# Peer review of "Adaptive evolution at mRNA editing sites in soft-bodied cephalopods"

_PeerJ, doi:10.7717/peerj.10456_

## Round 0.1 · original submission · Minor Revisions

Both the reviewers and I found this paper to be highly interesting, very well written, including a broad overview summarizing recent knowledge related to RNA editing, while suggesting a novel, scientifically-sound theoretical framework for A-to-G substitution at editing sites. This framework might have an important evolutionary role, which should be further investigated in future research.

Reviewer #1 had a few minor comments that should be addressed before fully accepting the paper.

·

Basic reporting

no comment

Experimental design

no comment

Validity of the findings

no comment

Additional comments

The manuscript by Moldovan et al. studies mRNA editing sites (A-to-I) using transcriptome data from 6 species of cephalopods. The authors provide conclusive evidence for positive selection of the A-to-G substitution at editing sites, consistent with a recent population genetic study in Drosophila and humans (Popitsch et al. 2020) that found similar signals. Intriguingly, this suggests that A-to-I editing could have conserved evolutionary roles in multiple diverged metazoan lineages.

The manuscript is excellently written, with a very clear overview of possible evolutionary models for RNA editing. They further suggest a novel and from my perspective very reasonable theoretical framework for the observed signals (i.e., non-functional editing sites enhancing the expressed genetic variability in species with low genetic diversity). The statistical and evolutionary analysis is done very thoughtfully and convincingly. I highly recommend acceptance of this study at PeerJ.

See some very minor suggestions below:

1) Line 204: EL is used here for the first time, write it out.

2) Line 454: It's not a very good choice to use "N" here as the number of possible editing-enhancing mutations since N is usually reserved for population size.

3) It would be interesting to quantitatively know to what degree the editing level can be predicted by sequence context and RNA structure, e.g. with multiple regression. This would make the biological significance of these factors clearer.

4) Line 154: It is difficult to understand R based on the description here (although after reading the results it became clearer). A numerical example could help (i.e. what are the numbers in the contingency table of the respective chi-square test). Same for the definitions below starting at line 172.

Reviewer 2 ·

Basic reporting

The manuscript is clear and written well. The introduction clearly lays out the background with all the relevant and recent references. The fundamental purpose of A to I RNA editing, why and how it arose over the course of evolution still very much eludes us. This paper takes a novel approach, using well founded data from previous work, to try and further elucidate this process. The use of Cephalopod editing data for such a study is clearly optimal due to the extensive editing events they exhibit. The additional discussion in the supplement regarding genetic assimilation and preadaptation is very important and adds to the background.

Experimental design

I am afraid that I lack the appropriate mathematical background to properly assess the calculations used, however they appear to be in order and are based on a well founded literature. That said, as mentioned above, this study is well within the scope and interest of PeerJ. The research question is well defined and the data used is quite appropriate. This manuscript certainly identifies and then fills in an important knowledge gap in this field.

Validity of the findings

The similarity of the findings from this study to those in Popitsch et al., 2020 for Drosophila and humans are quite encouraging, particularly from a broader evolutionary perspective.
The data is provided, here or cited from the previous studies used (particularly - Liscovitch-Brauer et al., 2017). The findings, particularly with regard to the sequence context and structure and their respective association with editing follows along a well studied line and expands on it. The notion that the vast number of non synonymous editing sites may in fact be conductive to the acceleration of the evolutionary process are fascinating and well worth pursuing with further work yet are of course at this stage speculative.

---

## Round 0.2 · accepted · Accept

I have found that the revised manuscript properly addressed all the concerns and suggestions that were made by the reviewers, and therefore I am pleased to accept your manuscript for publication in PeerJ. Looking forward to seeing it published.